# Ciguatera Mini Review: 21st Century Environmental Challenges and the Interdisciplinary Research Efforts Rising to Meet Them

**DOI:** 10.3390/ijerph18063027

**Published:** 2021-03-15

**Authors:** Christopher R. Loeffler, Luciana Tartaglione, Miriam Friedemann, Astrid Spielmeyer, Oliver Kappenstein, Dorina Bodi

**Affiliations:** 1National Reference Laboratory of Marine Biotoxins, Department Safety in the Food Chain, German Federal Institute for Risk Assessment, Max-Dohrn-Str. 8-10, 10589 Berlin, Germany; Astrid.Spielmeyer@bfr.bund.de (A.S.); Oliver.Kappenstein@bfr.bund.de (O.K.); Dorina.Bodi@bfr.bund.de (D.B.); 2Department of Pharmacy, School of Medicine and Surgery, University of Napoli Federico II, Via D. Montesano 49, 80131 Napoli, Italy; luciana.tartaglione@unina.it; 3CoNISMa—National Inter-University Consortium for Marine Sciences, Piazzale Flaminio 9, 00196 Rome, Italy; 4Department Exposure, German Federal Institute for Risk Assessment, Max-Dohrn-Str. 8-10, 10589 Berlin, Germany; Miriam.Friedemann@bfr.bund.de

**Keywords:** ciguatera poisoning, *Gambierdiscus*, fisheries management, ciguatoxin, harmful algae, marine toxins, human health

## Abstract

Globally, the livelihoods of over a billion people are affected by changes to marine ecosystems, both structurally and systematically. Resources and ecosystem services, provided by the marine environment, contribute nutrition, income, and health benefits for communities. One threat to these securities is ciguatera poisoning; worldwide, the most commonly reported non-bacterial seafood-related illness. Ciguatera is caused by the consumption of (primarily) finfish contaminated with ciguatoxins, potent neurotoxins produced by benthic single-cell microalgae. When consumed, ciguatoxins are biotransformed and can bioaccumulate throughout the food-web via complex pathways. Ciguatera-derived food insecurity is particularly extreme for small island-nations, where fear of intoxication can lead to fishing restrictions by region, species, or size. Exacerbating these complexities are anthropogenic or natural changes occurring in global marine habitats, e.g., climate change, greenhouse-gas induced physical oceanic changes, overfishing, invasive species, and even the international seafood trade. Here we provide an overview of the challenges and opportunities of the 21st century regarding the many facets of ciguatera, including the complex nature of this illness, the biological/environmental factors affecting the causative organisms, their toxins, vectors, detection methods, human-health oriented responses, and ultimately an outlook towards the future. Ciguatera research efforts face many social and environmental challenges this century. However, several future-oriented goals are within reach, including digital solutions for seafood supply chains, identifying novel compounds and methods with the potential for advanced diagnostics, treatments, and prediction capabilities. The advances described herein provide confidence that the tools are now available to answer many of the remaining questions surrounding ciguatera and therefore protection measures can become more accurate and routine.

## 1. Introduction

The identified risks, impacts, and challenges faced in the 21st century are numerous, encompassing economic, geopolitical, societal, technological, and environmental pressures [1]. Worldwide, oceans are enduring anthropogenically induced hazards and impacts including changes to physical conditions (e.g., nutrient loads [2], ocean pH, and warming waters [3]), ecosystem wide disturbances, and disruptions to ecosystem services and the fisheries industry, as highlighted by the Intergovernmental Panel on Climate Change (Figure 1) [4]. Any reduction in the supply of seafood products would negatively impact the livelihoods and primary source of animal protein for over one billion people around the world [5]. Signs of biological or ecological disturbances that threaten seafood security can include: (a) loss of marine biodiversity (including the loss of rare but functionally important organisms), (b) lower biomass (community or species), (c) loss of habitat, (d) the spreading of invasive species, or (e) increases in harmful algae and hypoxic dead zones. Small island developing states and their artisanal fisheries are particularly vulnerable to these seafood insecurities and are dependent upon a secure local seafood supply [6,7]. Otherwise, they risk becoming increasingly dependent on imported foods and foreign assistance programs [6,8,9]. Therefore, it is a major stated goal of the Food and Agriculture Organization of the United Nations to increase the contribution of small-scale fisheries to alleviate poverty and food insecurity [6,8].

A major impediment to this goal is a serious health threatening issue called ciguatera, responsible for significant harvest restrictions to avoid primary health risk-associated products [10,11,12,13,14]. Ciguatera poisoning (CP), often referred to simply as ‘ciguatera’, is the most commonly reported seafood toxin-related illness, endemic to many tropical and subtropical regions around the world and caused by the ingestion of seafood contaminated with potent neurotoxins, mainly ciguatoxins (CTXs) [11,15,16]. In general, shallow, warm water, marine habitats (<200 m, down to <1% of ambient surface irradiance, >16 °C, and >15 salinity) can support a population of benthic dinoflagellates in the genera *Gambierdiscus* and *Fukuyoa* [17,18]. Species in these genera can produce CTXs [19,20], and/or other bioactive ladder-shaped polyether compounds; such as gambierol [21], gambieroxide [22], gambieric acids [23], gambierone [24], 44-methyl gambierone [25,26,27], or maitotoxins (MTXs) [26,28,29,30,31,32], as summarized by the FAO and WHO [16]. Primary consumers (herbivores, planktivores, detritivores, omnivores, and possibly zooplankton) directly or indirectly consume *Gambierdiscus* and/or *Fukuyoa*, inadvertently acquiring their toxic compounds or toxins. CTXs (and potentially MTXs [33]), in particular, can bioaccumulate and become biotransformed in the consumer and move throughout the food-web via complex poorly understood predator/prey pathways, ultimately reaching humans when these toxin containing animals are consumed [25,33,34,35,36,37,38,39].

CTXs (CTX1B LD_50_ = 0.25 µg/kg, mice, i.p. [40]), the presumed causative agent of CP outbreaks, and MTXs (MTX-1 LD_50_ = 50 ng/kg, mice, i.p. [28]), are among the most toxic natural substances known and no specific antidotes for the illnesses caused by them exist [41,42]. CP occurrences by region can be found in Figure 2A [43] along with satellite-derived monthly average sea surface temperature ≥15 °C for the global peak summer (Figure 2B) and winter (Figure 2C) seasons of 2020. This set of images shows CP endemic areas and the territories with emerging incidences since the beginning of this century (Figure 2A), and where the most recent extent of thermal tolerance (Figure 2B,C) is likely to already be suitable for seasonal habitability for the species involved in CP. Where habitats become suitable for the presence of *Gambierdiscus* spp. and *Fukuyoa* spp., (here after *Gambierdiscus* and *Fukuyoa* unless specified) they likely signal the potential for future CP events where harvest areas or species considered ‘safe’ may transition to ‘emerging area/species of concern’ [44]. Ciguatera incidences are expected to increase, due to an accelerated expansion of the toxin-producing microalgae, which are based on forecasts and data-driven models of sea-level rise and warming waters [45,46] resulting from climate change and anthropogenic impacts [47,48,49,50,51,52,53,54].

The ciguatera-issue, as a whole, poses a problem for a multitude of specialists, from phycologists working on unraveling the many mysteries of microalgae to physicians and psychologists determined to help patients better understand and recover from the long-term neurological and psychological damage induced by CP [16,55]. Thus, ciguatera provides a strong case study demonstrating the importance of inter- and intra-agency multidisciplinary cooperative approaches for the transfer of shared working materials, training, and knowledge dissemination in order to advance the understanding of this complex phenomenon. From an environmental perspective on CP, there are many uncertainties to estimating the impact of biotic or abiotic, physical, spatial, and temporal changes to an ecosystem and food web regarding their role in dampening or exacerbating the CP issue. Both human and environmental factors are interdependent and further contribute to the severity and uncertainty around the impacts of ciguatera globally.


Figure 2Global map of endemic areas for ciguatera poisoning and satellite-derived sea surface temperatures (SST) divided into two scales, <15 °C and ≥15 °C. (**A**) map from https://www.ciguatera.pf/index.php/en/la-ciguatera/ciguatera-distribution (accessed on 15 March 2021) and (reproduced with permission from © Louis Malardé Institute 2020) showing † areas with indigenous ciguatera cases reported and Ф areas with indigenous ciguatera cases reported since 2000. (**B**) Satellite-derived average SST for the most recent peak month of summer in the northern (projection: https://go.nasa.gov/3m1rVhW (accessed on 15 March 2021)) and southern (projection: https://go.nasa.gov/35X8Vvp (accessed on 15 March 2021)) hemisphere, combined. (**C**) Satellite-derived average SST for the most recent peak month of winter in the northern (February 2020) and southern (August 2020) hemisphere, combined. Gray scale temperature map encompasses the temperatures <15 °C. Red scale color palette is from 15 to ≥32 °C. Satellite SST data generated from the NASA Worldview National Atmospheric and Space Agency’s EOSDIS Worldview app, version 3.8.2. Geostationary imagery layers generated from the following sources [56,57,58,59].
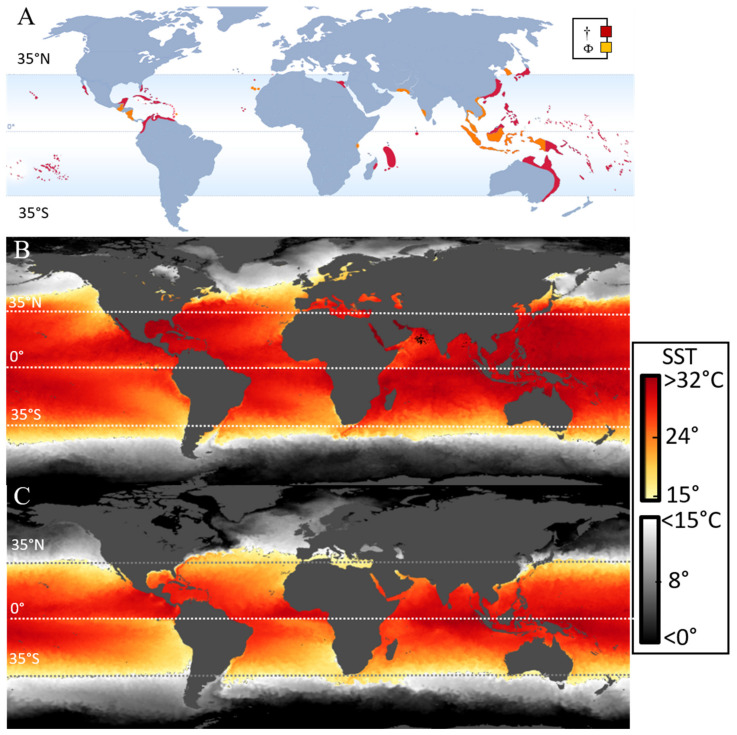



The burden of identifying the problem, mitigating the risk, and creating a strategy for reducing the incidence of ciguatera primarily falls upon resource and human health managers [10,16,60,61,62]. Numerous countries combating CP around the world have implemented domestic policies, like regulating the sale or capture of products implicated in poisonings, such as fish, by size, species, and region, with varying degrees of success [10,12,14,60,63,64,65]. However, domestic solutions are insufficient for this transnational global seafood-related human health problem. A problem that necessitates a multi-disciplinary effort involving scientists, consumers, governments, human health experts, and fishers; working together with a dedicated approach for capacity building, willingness to provide technology and knowledge transfer and to share research material to truly address the remaining challenges of CP [11,16,66,67,68,69,70,71].

### 1.1. Toxin Production

Currently, 18 species of *Gambierdiscus* and 3 of *Fukuyoa* are described [16,72], exhibiting differences (within- and among-species) in toxicity and compound production [29,73,74,75,76,77,78,79,80,81]. Of these, only a few species (i.e., *F. paulensis* [82], *G. australes* [83], *G. toxicus* [19,83,84], *G. polynesiensis* [20,78,81,85]) have thus far been documented to produce significant amounts of CTXs (CTX3B, -3C, -4A, -4B, 2-OH-CTX3C, M-Seco-CTX3C). Other species capable of producing CTXs have been suggested, but not confirmed. The species identification of collected microalgae isolates was accomplished using phycological techniques and targeted molecular approaches (e.g., fluorescence in-situ hybridization probes, restriction fragment length polymorphism typing, high throughput sequencing metabarcoding, or polymerase chain reaction) [79,86,87,88,89,90,91,92,93,94,95,96,97]. Field and culture studies identified several factors influencing cell abundance and species distribution, both spatially and temporally. These factors include physical oceanographic properties (e.g., irradiance, wave power, depth, salinity, and temperature), biological (predation, direct and indirect disturbances), and anthropogenic drivers (habitat modification, pollution, and fishing practices) [10,15,98,99,100,101,102,103]. In culture, *Gambierdiscus* has displayed inter- and intra-species variability for growth, morphology, and quali-quantitative toxin profile [83,104], which can be influenced by capture location, culture parameters, and time spent in culture (phycological drift) [20,73,74,75,104,105,106,107,108]. Interactions between bacterial or other microbial co-inhabitants and *Gambierdiscus* indicated antagonism, competition, or benefits with impacts on fitness and bi-directional regulation in compound production [109,110,111]. From the perspective of the algae, it remains undescribed whether CTXs, MTXs, and other known compounds (e.g., gambierone, gambieric acids) are produced for inter- or intra-species defense or benefit. Due to the variability within and among species, estimating the toxin content (quantity or type) of a given area based on a knowledge of microalgal species presence/absence/ratio or abundance, without knowing if in-situ cell counts translate into higher toxin ingestion rates of fish, renders seafood safety prediction, based on cell counts, unreliable (at present).

Historically, there were three families of CTXs described, whose chemical structure was labeled loosely based on their geographic origin, namely Caribbean Sea (C-CTXs), Pacific Ocean (P-CTXs), and Indian Ocean (I-CTXs). Although these region-specific descriptions are still in use, the FAO and WHO ‘Report of the expert meeting on ciguatera poisoning’ recommends a description based upon known chemical structures and geographical distribution. For this adapted approach CTXs, and their derivatives (which include >30 analogues) were classified into four separate groups: CTX4A, CTX3C, C-CTX (Caribbean ciguatoxin), I-CTX (Indian ciguatoxin) as summarized by the FAO & WHO [16], and the references therein. Several questions regarding the structural elucidation of I-CTXs and the microalgal origins of I- and C-CTXs remain unresolved. Toxicity equivalency was investigated for successfully isolated CTXs (of variable purity); however, only CTX3C and CTX1B are commercially available (manufacturer: FUJIFILM Wako Chemicals Europe GmbH (Europe Office) Neuss, Germany, as of February 2021). Therefore, as new commercial sources of standards become available, from certified producers with verified purity, accurate toxicity equivalency should be reassessed [52,112,113,114].

### 1.2. CTX Vectors

Trophically, microalgae are situated at the base of the food web. When the algae are consumed, their toxin contents can be biotransformed and can bioaccumulate in complex multi-directional pathways throughout the food web [36,41,101,115,116,117]. Therefore, with improvements in detection sensitivity and better insights, it is likely to be proven that most (if not all) species feeding in hyperendemic CP regions accumulate CTXs, at some point in their life history to some degree. Consumer-resource interactions are a core factor in the transfer of CTXs, where different feeding mechanisms including detritivores, herbivores, planktivores, omnivores, invertivores, and piscivores are involved in the bioaccumulation and transfer of CTXs. CTX profiles in animals are region and species-specific, reflecting a behavior or consumption pattern descriptive of how they acquire CTXs [35,36,118]. The modification of CTXs from the less oxidized algal metabolites (e.g., CTX3C, CTX4A/4B) into the highly oxidized (and higher potency) analogs found in fish at higher trophic levels (e.g., CTX4A/B into CTX1B), are associated with (and potentially require) a pathway-specific to animal metabolism or metabolic transformation [41]. This pathway is not always linear, but generally, fish and shellfish that feed on low trophic level species are associated with containing mainly the algal specific compounds (the less oxidized forms/lower toxic potency) [35,119]. Each species individual traits (e.g., home range/distances traveled, feeding mechanism [120], size, prey selection/hunting behavior [121], lifecycle [122]), and even the prey available in the season [123], determine where and how fish feed and, consequently, influence diet and ultimately fish’s CTX profile [124,125]. Therefore, the feeding mechanisms and unique traits for each species present an opportunity to correlate CTX congener analysis with species, behaviors, or regions (similar to using naturally occurring stable isotopes to answer many of these environmental/nutritional/biological/ecological questions). For example, many species begin life in a nursery-type habitat (e.g., mangroves or seagrass beds), where they hunt small prey (e.g., crustaceans and plankton) [126]. As fish grow and mature, they often undergo an ontogenetic diet shift moving towards locations or habitats (e.g., deep shelf, coral reef) where they may source food that can have a higher total body CTX content (i.e., size as an indicator for prey selection). Inter-annual or seasonal spawning areas and migration routes can contain staging areas, courtship areas, or temporary locations to source food where CTX accumulation can occur [127,128,129,130]. These mixed-use and seasonal movements are becoming better understood, but can still lead to complications for historical CTX predictions and associations. CTX contamination seemingly occurs in food webs sporadically [44,131,132]. Not all fish within a given reef or common catch area contain CTXs, and fish caught shoaling or schooling together can vary greatly in their CTX content, creating temporal and spatial prediction uncertainties [10,44,64,131,133].

After ingestion, CTXs are metabolized in different ways, subject to the animal’s individual biochemical/biological/physiological processes. Generally, CTXs are taken up and first detected in the gastrointestinal tract and liver, then eventually transferred to the muscle tissue and throughout the body and skin [134,135,136,137]. Studies examining the behavior of fish administered CTXs found both behavioral abnormalities and the absence of impacts [135,137]. The application of next-generation sequencing in fish ecotoxicogenomics could help bridge the link between exposure and effects, a benefit for environmental assessments for CTXs as well as assessing risks to fish health [138,139]. Available data regarding the absorption efficiency of CTXs from fish feed was approximately 1–6%, where depuration half-life rates were tissue-related, from several days for the liver to several months in the skin [134]. Within the fish reside many useful biological data points for gaining insight into CTX accumulation, including age (through otolith analysis), dietary shifts due to prey availability or ontogeny (stable isotopic analysis to determine trophic feeding level, location, or habitat utilization type), fish prey information (analysis of gut contents) [140], fish size (by standard morphometrics), fecundity by egg production/weight (impact of CTXs on fish reproduction), and the liver/viscera/brain for bioaccumulation and metabolism data. Correlating these data to the measured CTX burden ultimately provides insights into CP. Future monitoring efforts for these data sources could provide past (CTXs/skin), present (CTXs/flesh), and future (CTXs/liver) temporal stages of CTX progression through the localized food web, together with data obtained for *Gambierdiscus* (CTXs/cell, cells/area). Testing vectors of CTXs, while understanding their behavior, lifecycle, and movement patterns, will help to provide insight into the trophic transfer of CTXs and potentially constrain areas of CP concern, spatially and temporally, to protect consumers.

Some species are locally considered ‘safe for consumption’ [141]. However, species implicated in CP can have wide geographical ranges, e.g., the Nassau grouper (*Epinephelus striatus*) transcend the jurisdictions of 43 nations or territories [142], or have a relatively small spatial home and hunting ranges, with occasional sporadic movements for spawning aggregations and migrations. Consumer avoidance of CP risk species can become complicated when identifiable characteristics used for phenotypically determining a species, like the head or skin, are removed during processing [143,144] or when species hybridize [145]. Therefore, the policies guiding CP harvest restrictions can be subverted by deliberate or inadvertent species substitution [146], or by falsifying capture location information [147]. However, sole reliance on previous experience with CP, without updated information, can also be problematic, particularly when climate change is projected to affect the behavior and life cycles of fish leading to new and potentially unforeseen consequences [148,149,150,151,152].

The world’s oceans have absorbed heat and CO_2_ associated with climate change. Excess heat affects the oxygen content of water, which in turn increases the metabolic costs associated with breathing, resulting in changes to behavior, metabolism, respiration, body size, and the life history of fish [152,153,154,155,156,157,158]. In the US Virgin Islands (a hyperendemic region for CP), herbivorous fish were observed taking 20,000–156,000 bites m^2^ d^−1^ [159]. *Gambierdiscus* abundance on algae can range from 1-1,000,000 cells g^−1^ wet weight of algae [160] and toxin content per cell can exceed a 1000-fold difference between the least and most toxic species [74]. Therefore, any increase in metabolism, due to a lower oxygen content or higher temperature, could result in a higher cell (toxin) ingestion rate. Rising CO_2_ concentration in the ocean has resulted in ocean acidification (Figure 1). Calcium carbonate dissolves in acidic conditions, therefore a lower pH in the ocean will impede the calcification process for shell-forming organisms (e.g., shellfish, corals), and even have an impact on fish behavior. A state shift in the coral ecosystem via loss of coral or shellfish bed structural habitat would benefit turf or macroalgae, a preferred habitat for CP producing microalgae and food source for herbivores. Additionally, climate influences the movement of animals [125,161,162], and better insight into the spaces utilized, movement, and connectivity [163] is desired to make informed decisions concerning how these changes impact the required risk management of CP and subsequent policies.

### 1.3. Human Health: Ciguatera Diagnosis, Epidemiology, and Traceback Investigation

Challenges to fully understanding the CP impacts on humans are numerous and include: (i) difficulties in identification and misdiagnosis due to diverse unspecific symptomology (e.g., gastrointestinal, neurologic, and cardiac) [143], (ii) severity related to consumption habits (what parts of the animal were eaten and how much) [114], (iii) patient pre-existing conditions or existing CTX body burden, (iv) willingness to seek treatment [164,165], (v) whether the healthcare system considers ciguatera to be a reportable illness [11,166], (vi) globalized intertwined food networks [146,167], and (vii) international trade and travel where the source of illness (traceback) can be difficult to ascertain [144,168]. In most CP incidents, symptoms are mild or self-resolving, a frequently cited reason for underreporting issues related to ciguatera. CTXs have bioaccumulative properties, therefore, the human exposure at low levels over time may represent a potential human health risk and requires further elucidation [169,170,171]. Furthermore, where ciguatera cases are rare or unheard-of, awareness of the problem is low, even among healthcare workers [172]. Misdiagnosis can be common and clinical registration with the International Classification of Diseases (ICD-10) under code T61.0: ’Ciguatera fish poisoning’ or reporting CP cases with emergency department databases, public health, or data curation authorities is not always mandatory. Information from patients, including consumed species (amount and parts of fish) and patient personal information (e.g., weight/age/health), are important data for risk assessors, epidemiologists, and managers [11]. Recent advancements in CTX detection efforts in biological samples (blood, urine) provide a clear path towards independent laboratory confirmation of the clinical diagnosis in addition to meal remnant analyses [173,174].

In the early 21st century, there has been a strong consumer trend towards ethical/health-conscious and sustainable diets, based on alternatives to meat-based protein (e.g., vegetarian, vegan, plant-based), mainly due to environmental, health, religious, philosophical, and ethical reasons [175,176,177,178]. This outlook on protein source consumption has relevance for improving the accuracy of CP incidence rates, where consumers who abstain from the consumption of fish and fish-based products (entirely or based on region or species) could be excluded or refined for risk assessments, presenting a more accurate representation of the risk group [64,165]. For example, in an area of emerging risk for CP, the United States Florida Department of Health reported an unadjusted annual incidence rate of 0.2 reports of CP per 100,000 people, while after adjustment for underreporting and a focus on groups of high CP risk the number was projected to be 2000 times higher (400 per 100,000) [179].

Precautionary consumer protection, through the prevention of hazardous contaminants in food products from reaching the consumer market, is a priority for health and food safety organizations, and these efforts include prevention of CP. The European Union’s fisheries and aquaculture products market is among the most valuable worldwide and fish with CTX-group toxins are forbidden from this market [180,181]. However, currently, there is no reliable, cost-effective, commercially available preemptive fish-testing commodity for CTXs [182] to ensure a product is CTX free before reaching the market. Additionally, CTXs and MTXs are undetectable by organoleptic methods and are relatively unaffected by acidic and thermal conditions, limiting typical consumer self-protection food preparation options if a product with CTXs is purchased [183,184]. A diagnosis of CP is currently based on a recent history of eating fish, the clinical presentation of CP, and if possible analytical testing of a meal remnant, or if a portion is unavailable, a related lot [185]. So far, a meal remnant analysis remains the best scenario for confirmation of potential poisoning. In trace back instances from CP outbreaks, DNA-barcoding is a critical tool for the identification of the ciguatera-causing species [145,186,187,188,189,190,191,192,193]. Toxin identification, quantification, and DNA barcoding [194] can still be performed on the meal remnant even after cooking, yielding valuable information for cataloging CP events and causative fish species [114,186,195].

Historically, in an artisanal (or traditional/subsistence) fishery, products were consumed close to where they were captured, and anglers sold directly to consumers, incentivizing them to ensure safety and sustainable harvest of the product [132,133,196]. With a globalized food distribution network, advanced food processing and preserving techniques, it can be difficult to ascertain the product’s history, and any inherent potential risks as these products have a much longer shelf life and no restrictions on distance for distribution. Modern advancements have brought digital solutions to value (product) supply chains to meet these uncertainties, from a supplier, consumer, social, and environmental perspective. For example, efforts to increase fish stock knowledge, seafood traceability and transparency can benefit from novel approaches such as Blockchain [197], Radio Frequency Identification Device tags for product authentication (including species and catch data), applications of machine learning [198], data mining, artificial intelligence, and other digital technologies [199,200,201]. These methods of digitally-enabled food supply chains can help promote sustainable development goals within the context of the fishing industry, which aims to reward responsible and ethical producers and to discourage illegal or unethically produced seafood products from entering the seafood supply chain [202]. Eventually, as the traceback methods based on these technologies become routine and are coupled with modern testing efforts, these understandings and efforts will begin to support one another. Together they will help accelerate the capabilities to predict: (i) when/where fish containing CTXs are originating from (ii) how/why these products came to contain CTXs, (iii) what CTXs or analogues they contain, specifically in terms of helping to identify *Gambierdiscus*/*Fukuyoa* or specific regional type toxin profile. Therefore, while these digital technologies are still being implemented, to improve and support the accuracy of epidemiological data regarding CP vectors and their geographic origin, collection of the location, species, and morphological data via all available methods and sources is desired.

### 1.4. Modern Testing and Investigation Capabilities

The FDA established guidance levels of 0.10 and 0.01 ppb equivalent toxicity in fish for C-CTX1 and CTX1B (CTX4A derivative), respectively. The detection of CTXs at these concentrations in complex matrices, with recoveries between 50–100% requires appropriate facilities and advanced analytical methodologies [10,60,203,204,205,206,207,208]. Extraction procedures before testing are variable and often time consuming (up to 3 days). Currently, no chemical protocol for CTX extraction in a biological matrix has been validated, an analytical challenge discussed in the FAO and WHO report [16]. The CTX testing protocols include bioassays (in-vivo and in-vitro), biochemical assays, and chemical assays, each selected based on the user’s capabilities, requirements, or intentions [16,209]. Those used for supporting the clinical diagnosis of ciguatera (when a meal remnant is available) include the species identification of a meal remnant through DNA barcoding and the toxin analysis using a two-tiered approach (e.g., The U.S. Food and Drug Administration’s (FDA) method outlined in [11,186,210]). Generally, toxin analysis includes (i) a semi-quantitative screening method capable of measuring composite toxicity in an action-specific or dose-response manner [210,211,212,213,214,215,216,217,218,219] and (ii) confirmation of CTXs identity either by liquid chromatography tandem-mass spectrometry (LC-MS/MS) or LC coupled to High-Resolution MS (LC-HRMS) [36,85,113,183,186,205,210,220,221,222,223]. However, many different methods (including local folk methods [61,224]) were implemented for the detection of CTXs, as recently reviewed by Pasinszki et al. [225]. Beyond the clinical analysis support, the two-tiered methods/protocols (semi-quantification paired with confirmation) are commonly used for the monitoring of CP risk in the frame of surveillance programs for sample analysis of micro-algae, fish, marine invertebrates, solid-phase adsorption toxin tracking filters, or other artificial surfaces/material with the potential to contain *Gambierdiscus/Fukuyoa* or CTXs, which have been collected in areas of interest. Currently, no validated or accredited method exists for the routine analysis of CTXs.

Methods employed for the first tier ‘screening type’ approach improved reliability and sensitivity, and are capable of detecting toxins below the guidance levels. Historically, in-vivo-based methods, while not specific, were widely used for toxicity screening, however, due to various analytical and ethical concerns, alternative methods are preferred [181,226]. Most in-vitro-based methods, designed to investigate the neurocellular effects of CTXs using various cell types (mouse, guinea pig, rat, human, etc., as reviewed by L’Herondelle et al. [227], and references therein), require the use of a protein-based serum and the use of essential supplements in cell culture. These supplements are mainly animal-based (e.g., fetal bovine serum), which also raises the scientific reliability/repeatability and animal-based ethical concerns of in-vivo methods. Therefore, the design of cell lines that are serum-free/reduced [228] and suitable for a variety of applications are desired; and currently available for some cell lines and in certain instances [229,230,231,232]. Further cell line modifications to improve the sensitivity/specificity/ease of use/reliability could benefit from the advent of Clustered Regularly Interspaced Short Palindromic Repeats (CRISPR), CRISPR-associated protein (Cas9), and its subsequent refinements, which can now make genome editing for a better understanding of the genetic basis for observations possible [233,234,235,236,237,238,239]. This enables the potential to modify the genome of a cell line, or even algal culture, utilizing a loss-of-function or gain-of-function mutation. For example, this technique has been/can and will be used for genes that encode the various voltage-gated sodium or potassium channels (e.g., knocking-out, increasing, or reducing channel function to tailor a cell line for a specific mode of action or sensitivity). Furthermore, those techniques can be used for the targeting of genes that can up- or down-regulate the toxin (or another compound of choice) production of cultured microalgae or of genes that help to adapt a species to survive the conditions suitable for culture (allowing further investigations). This can also become an important tool for investigating marine biotoxins (e.g., CTXs, brevetoxins, and tetrodotoxins), which have variable affinities to a variety of voltage-gated sodium channels that require further elucidation [240,241,242,243,244,245,246].

Other efforts to understand the function and processes of cells can benefit from various genomic, proteomic, metabolomic, transcriptomic, and glycomic approaches (OMIC’s) to elucidate a diverse array of cellular processes with relevance for drug discovery, food safety [247], method development, and toxins research [248,249,250,251]. Recent advances in the development of monoclonal antibodies, specific against CTX1B, 54-deoxyCTX1B, CTX3C, and 51-hydroxyCTX3C have been achieved [252,253,254]. The availability of monoclonal antibodies was instrumental in the development of sensitive enzyme-linked immunosorbent assays [214,215,255,256,257], magnetic bead-based immunoassays, and immunosensors [216,258,259]. Specific and robust antibodies are needed to develop lateral-flow assays, which are simple, low cost, and with field deployment potential [260]. The design of a lateral-flow assay can be tailored for the analysis of cells, toxins, or genetic material [261]. Additional immunosensing tools (colorimetric immunoassay and electrochemical immunosensors) are available to screen for specific CTXs and genetic information for species identification [216,258]. For observing and measuring cells, microscopy presents a tool commonly used that remained relatively unchanged for decades, however, digitalization has led to rapidly increasing imaging resolution and portability (two desired performance demands), even moving away from refractive lenses in conventional optical settings, enabling ultra-thin, ultralight, and flat imaging systems [260]. Super-resolution imaging of structures in three dimensions at the nanometer-scale [262,263] and high-resolution live cell Raman images can further elucidate many of the questions regarding marine toxin site binding, affinity, and intercellular impacts [264,265].

Among analytical techniques suitable for tier two ‘confirmation’, LC-MS/MS is the most utilized for the qualitative (and at times quantitative) determination of CTX analogs due to its capacity for high selectivity and sensitivity in detecting analytes at sub-ppb levels in complex matrices [204,225]. The full implementation of LC-MS methods requires sufficient materials (toxins in natural matrices, reference materials, and analytical standards): (i) to optimize analyte ionization and fragmentation settings, to increase sensitivity for reaching the sub-ppb detection levels required, (ii) to chromatographically resolve different structural analogs (of which there are >30), and (iii) to test the applicability of the method on naturally contaminated and/or spiked samples for method validation (a point that also applies to the tier-one screening methods). Challenges for LC-MS method development stem from the laboratory’s LC-MS instrument platforms utilized, including different electrospray ionization source geometries, which can influence the ionization behavior and ultimately detection. The vision of transportable, handheld, miniaturized mass spectrometers for on-site food safety testing is becoming a reality for a variety of applications, with a wide range of molecular weights and polarities being demonstrated [266,267,268]. However, regardless of the method type, the availability of CTX certified standards or reference materials remains a crucial impediment to method development. Several CTX reference materials have been reported in the literature, generated from naturally incurred material, however, their availability is private and shared among researchers only as gifts or private purchases. Only two CTX congeners, CTX3C and CTX1B, are now commercially available (manufacturer: FUJIFILM Wako Chemicals Europe GmbH (Europe Office) Neuss, Germany, as of February 2021). The availability of authentic analytical standards will aid the improvement of detection and sensitivity capabilities [269,270], which will continue to benefit consumers at risk of chronic low dose exposure to CTXs with the final aim of the inter-laboratory validation of the MS-based methods [271]. This goal of inter-laboratory validation also applies to tier-one screening methods (CBA-N2a, RBA, ELISA, etc.). However, all methods are subject to uncertainties, e.g., insufficient cleanup, matrix suppression, method execution errors, misinterpreted data. Therefore, improving selectivity, sensitivity, repeatability, and reproducibility for CTX quantification in methods is necessary. Together these established, improved, and novel methods provide a wide range of options suitable for a method to detect CTXs (and MTXs), with varying degrees of difficulty for implementation. Therefore, different laboratories with a range of capabilities and infrastructure can select the screening type method sufficient for their infrastructure and purpose.

### 1.5. Anthropogenic Impacts

Humans are (inadvertently) exerting influence on the production and distribution of CTXs through changes to the presence, absence, and abundance of toxin-producing species, their vectors, and habitat. Changes in land use and hydrology (impervious surfaces, breakwaters, clearing vegetation, sediment, and freshwater runoff), poor water quality (pollution, hypoxic zones due to excess nutrients N, P, K), benthic habitat coverage, climate change, overfishing, and the introduction of invasive species can all alter food-web dynamics and CTXs in directions that are uncertain and can lead to non-linear impacts. Ocean warming during the last century has enabled tropical species, including *Gambierdiscus*, to expand their range poleward [48,272]. A rise in global sea level (approx. 1.7–3.2 mm yr^−1^) is converting landmass into new shallow water habitat, with a significant acceleration in the rate of sea-level rise predicted for the 21st century [273,274]. *Gambierdiscus* are forecast to benefit from these environmental changes [51,275,276], as they can tolerate a wide range of light intensities and depths (<1 to 150 m) [17]. Efforts to predict these impacts can utilize elevation-based assessment models that show digitally where newly submerged areas are likely to provide additional habitat to colonize for adjacent populations of *Gambierdiscus* in CP endemic areas [98,132,277]. Benthic habitat alterations due to natural and anthropogenic causes (e.g., coral bleaching [278,279], reef submersion due to sea-level rise, dredging, boat anchor scarring, loss of mangrove coverage) can alter the wave dampening and storm surge protection provided by natural habitats, further eroding the shoreline for oceanic submersion and creating newly available habitat for *Gambierdiscus* to colonize [280,281,282,283]. Common structural habitats for *Gambierdiscus* are macro- and turf algae, and changes in benthic habitat type from coral to algae [284,285,286] (e.g., decrease in ocean pH), or newly submerged habitat converted to macro- and turf algae cover can enhance the proliferation of *Gambierdiscus* [99,287,288,289,290]. A healthy grazer community keeps algal growth in check and excluding grazers (either by overfishing or physical exclusion) can throw off this balance and lead to an increase in *Gambierdiscus* abundance [98,291,292]. As oceans absorb more CO_2_, this may benefit algae and seagrasses, as carbon dioxide is required for photosynthesis aiding algal growth rates. *Gambierdiscus* are historically/currently region-specific [160,293] and the toxins they produce are used as regional profiles or biomarkers helping to identify the ocean of origin when investigating meal remnant samples from CP outbreaks. If these region-specific algae are transferred to non-native habitats (e.g., via marine litter [294,295], ballast water [296], accidental introduction) detection methods using the region-specific CTX biomarkers would become invalid as a traceback tool [36], concurrent with human health consequences [36]. Symptoms of CP originating in the region of the Pacific Ocean are primarily described as neurological, while gastrointestinal symptoms dominate the description in the Atlantic/Caribbean [183,297,298,299,300,301,302]. Therefore, if these species become invasive the region-specific symptomology may become unreliable. In culture, some *Gambierdiscus* species can respond to an environmental shock (e.g., sudden drop in temperature) by increasing toxin production [303] or loss in toxin production when transferred to a new light environment [304]. When species are introduced to a new environment (e.g., with seasonal cold upwelling), the exposure to novel conditions could result in behavior and responses that are uncharacteristic or unpredictable. Warming waters, size-selective fishing, and overfishing can alter the size structure of fish stocks toward smaller individuals, leading to ecosystem impacts that are diverse and extensive [305]. A decrease in length affects species interactions, biomass, fisheries yield (affecting food security), and ultimately results in unknown (and difficult to track) changes to food web pathways for CTXs [306,307]. The introduction of invasive species can elicit sweeping changes to food web dynamics [308,309,310], benthic cover [311], and invasive fish predators can become novel or dominant CTX vectors and ultimately CP sources [312,313,314,315].

Habitat loss, overfishing, invasive species competition, or long-term changes such as ocean acidification and climate change are serious anthropogenic threats to reef fish populations and the overall health of reef ecosystems. Their subsequent short and long-term impacts on CP incidences remain unknown. A pragmatic approach currently employed to mitigate human impacts on marine systems and to increase fisheries productivity [316] is to manage the fishery through various efforts such as the establishment of marine protected areas, or ‘no-take’ zones [317,318,319]. Harvest restrictions and marine reserves can also be a useful tool for protecting consumers from the consumption of CTX contaminated fish. However, CTX distribution can be geospatially complex and harvesting at the ‘spillover’ borders of these protected areas (if they are not sufficiently large) [127,319,320] can potentially carry an increased CP risk [44,132].

Climate change and the global response [321] can have unintended impacts on harmful algal species. To lower greenhouse gas emissions, various studies proposed offshore wind farms for sustainable energy generation; however, *Gambierdiscus* have been shown to inhabit such novel shallow water structures as demonstrated for oil platforms [322]. Other marine structures created to protect the shoreline from sea-level rise and erosion (e.g., artificial breakwaters, seawalls) can serve as ‘artificial reefs’ attracting a variety of marine life, which in turn are targeted by fishing communities [323,324]. These artificial reef-type breakwaters and barriers are designed to alter the wave energy approaching the coast [325,326,327]. Calm, low hydrodynamic habitats can benefit *Gambierdiscus* [102], therefore, anthropogenic marine modifications can have several impacts on CP. For instance, they could serve as habitat enhancers for CTX producing species, either through shallow water habitat ‘stepping-stones’, novel hard bottom to adhere, or by reducing ocean wave energy, or serving as fish aggregates modifying where CTXs enter the consumer-resource system. Consequently, their impact on CP should be monitored.

Globally, communities in the 21st century have demanded and enacted large social and environmental changes over the ‘business-as-usual’. This includes the 1995 Kyoto Declaration and Plan of Action to respect the resources, food security, social and economic development of developing countries (in particular small-island developing states) [328]. Additionally, the Nagoya Protocol on access to genetic resources and the fair and equitable sharing of benefits arising from their utilization set forth a framework for protection, as a reaffirmation of the sovereign rights of states to protect their natural resources and traditional knowledge [329,330]. Therefore, keeping with this progress it is important for CP-related research efforts to join these goals by striving to meet these standards. Ethical and conservation implications for both human (local indigenous peoples) and ecological communities should be ingrained in the research and their outcomes [331], particularly through ensuring equal partnerships in research and data sharing with local communities, while striving to minimize the ecological impact and reduction in harm to animals during sample collection (when necessary) [332,333,334,335].

### 1.6. Outlook- Risks and Opportunities

Increased awareness and efforts studying the intricacies of *Gambierdiscus*, CTXs, and CP have led to discoveries at an ever-increasing frequency for each step of the ciguatera problem from the toxin sources to the patients. Improvements in methodology and CTX detection sensitivity are enabling the early discovery of *Gambierdiscus* and CTXs in locations at the forefront of their expansion, while cell abundance and toxin levels are low. These monitoring efforts are constantly being refined as new physical oceanographic data (e.g., newly launched Sentinel-6 Michael Freilich ocean tracking satellite replacing low-resolution with higher resolution satellite observations), experience from fisheries scientists, CTX method research, food web transmission, and epidemiology surveillance data become available [132,297,336,337]. Further improvements to sample collection precision for *Gambierdiscus* can also benefit from the adoption of successful protocols, outreach (educational material) [301], and coordination efforts currently in use for monitoring other harmful algae species (where appropriate) [69,338,339]. The combined efforts of testing accuracy (e.g., species specificity and abundance) and routine-analytics for CTXs will contribute to refined surveillance data accuracy. These improvements are required to ‘catch up’ to the technology available for developing environmental (remote) biosensors for sensing *Gambierdiscus* and CTXs *in situ*. The acquisition of accurate CP-relevant data in near real-time is invaluable information for the design and testing of environmental predictor models, which marine and terrestrial spatial planners rely on to make informed decisions for mitigating CP.

The structures of a variety of marine natural toxins are currently established; however, the biosynthesis of many of these compounds remains unknown. Understanding the natural assembly of these toxins can benefit many research fields, including the potential for producing drug candidates or enhanced toxin detection. For example, Brunson et al. [340] used a multidisciplinary approach, where a breakthrough in the understanding of the biosynthetic pathway led to the expression of the involved genes in bacteria and yeast, resulting in the production and elucidation of a domoic acid isomer. The chemical synthesis of CTXs [269,270,341,342,343,344], gambierol [345], gambieric acids [345,346,347], and MTXs remains challenging [348]. Harvesting of microalgal biomass and large-scale culturing for purification of CTXs and MTXs continues to be an important process towards providing the necessary toxin amounts for full structural characterization and production of (certified) reference materials [16]. With sufficient material available, novel therapeutic applications can be explored by the pharmaceutical industry for new drug development based on the unique affinity for CTXs on mammalian sodium channels [349,350] or MTXs for the study of calcium channel-dependent processes, innate immune response, and physiology of inflammatory effector cells [351]. Moreover, sufficient amounts of microalgal material would allow investigations into the presence of a variety of large molecules, middle molecular-weight compounds, and natural products for bioorganic/biochemical research [352].

## 2. Conclusions

The examples of interdisciplinary cooperation and continued efforts (e.g., EuroCigua project) outlined in this review will continue to drive the efforts for combating CP towards the prediction of ciguatera in humans and the generation and movement of CTXs in the environment. Understanding these processes from toxin generation and transmission to human health impacts will continue to create benefits for society, in often unforeseen ways, i.e., via environmental protections and informed fishing decisions, the generation and exploration of compounds for beneficial purposes, and ultimately for ensuring a more safe and secure food supply.

## Figures and Tables

**Figure 1 ijerph-18-03027-f001:**
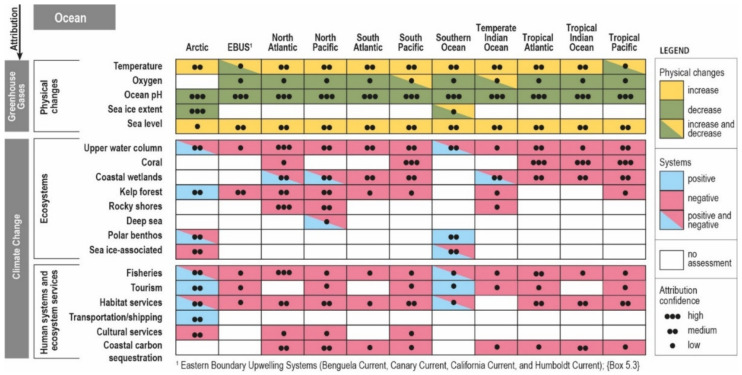
Synthesis of observed regional hazards and impacts in the ocean, assessed in the ‘Special report on the ocean and cryosphere in a changing climate’ (Figure 5.24 in the report). Detailed information about the legend and classification data are provided therein [4]. Many of these changes directly or indirectly influence the issue of ciguatera, as outlined in this manuscript.

## Data Availability

No new data were created or analyzed in this study. Data sharing is not applicable to this article.

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
