# Peer review of "Ciguatera Mini Review: 21st Century Environmental Challenges and the Interdisciplinary Research Efforts Rising to Meet Them"

_ijerph, 2021, doi:10.3390/ijerph18063027_

Round 1
Reviewer 1 Report
Dear Editor
In response to the evaluation of the paper "Ciguatera mini review: 21st century environmental challenges and the interdisciplinary research efforts rising to meet them"
First of all, congratulations to the authors, the article is well written and objective, with an interesting construction of the facts and a direct and concise approach to the problem.
I do recommend for publication without corrections
Att.
Author Response
Dear Reviewer 1,
Thank you for the review of our article, we appreciate the time you took to read it, needless to say, we were happy with your decision. Nonetheless, we have undertaken some minor changes for readability and corrected some errors found while reviewing. We hope you enjoy the finished product.
Actions: no changes requested.
Reviewer 2 Report
Abstract. The last sentence is very long and so difficult to read. Best split into two or three sentences.
Lines 72, 76, 91 and 92: Italicise Gambierdiscus and Fukuyoa.
L 77: ...acquiring their toxic compounds or toxins.
L 82: add reference to Munday's paper, your reference 37.
L131: ...and region, with ...
L207: ...locations, where fish may source food …...
L214: two possible references are
Murray JS, Harwood DT, Rhodes LL 2020 Ciguatera fish poisoning event in New Zealand from imported tropical reef fish and confirmation of Pacific ciguatoxins by LC-MS/MS. Harmful Algae News (IOC newsletter). 66: 4-5.
Murray JS, Harwood DT, Rhodes LL 2021. Ciguatera poisoning and confirmation of ciguatoxins in fish imported into New Zealand. The New Zealand Medical Journal (accepted).
L249: Murray et al. 2020
L255: herbivorous
L260: Awkward sentence. ...may include CTX metabolites....
L264: ..(figure 1). Calcium ….
L265: … and better insights ….
L293 and 320: Murray et al 200 and/or 2021 (just accepted).
L468: Include warming seas due to climate change? Reference:
Rhodes LL, Smith KF, Murray JS, Nishimura T, Finch SC 2020. Ciguatera fish poisoning: the risk from an Aotearoa/New Zealand perspective. Toxins 12: 50; doi:10.3390/toxins12010050.
Author Response
Daer, reviewer 2
Thank you for taking the time to review our article and to provide comments/suggestions. We have made the requsted adjustments and additions. A summary addressing each comment, point by point, can be found below. Thank you again for the efforts and insight.
Abstract. The last sentence is very long and so difficult to read. Best split into two or three sentences.
We have amended this sentence and broken it into three sentences for readability.
Lines 72, 76, 91 and 92: Italicise Gambierdiscus and Fukuyoa.
We have italicized all instances of the genus throughout the manuscript
L 77: ...acquiring their toxic compounds or toxins.
We have amended the sentence to reflect the requested changes.
L 82: add reference to Munday's paper, your reference 37.
Munday et al. 2017 is included in the sentence reference list.
L131: ...and region, with ...
The above mentioned edit has been made.
L207: ...locations, where fish may source food …...
We have amended the sentence to reflect the suggested changes.
L214: two possible references are
Murray JS, Harwood DT, Rhodes LL 2020 Ciguatera fish poisoning event in New Zealand from imported tropical reef fish and confirmation of Pacific ciguatoxins by LC-MS/MS. Harmful Algae News (IOC newsletter). 66: 4-5.
Murray JS, Harwood DT, Rhodes LL 2021. Ciguatera poisoning and confirmation of ciguatoxins in fish imported into New Zealand. The New Zealand Medical Journal (accepted).
Additional references have been added to the statement on Line 214. Unfortunately, while we look forward to the Murray 2021 article, it is not yet available to our institution and therefore cannot be read and unable to be added as a reference.
L249: Murray et al. 2020
While this article does discuss climate change and CP, line 249 is in reference to climate change impacts on fish behavior. Murray et al. 2020 was added elsewhere it was appropriate.
L255: herbivorous
This change has been made
L260: Awkward sentence. ...may include CTX metabolites....
This sentence has been adjusted and the suggested change has been made.
L264: ..(figure 1). Calcium ….
The suggested change has been made.
L265: … and better insights ….
The suggested addition has been made.
L293 and 320: Murray et al 200 and/or 2021 (just accepted).
Murray 2021 is unavailable, and Murray 2020 was added.
L468: Include warming seas due to climate change? Reference:
Rhodes LL, Smith KF, Murray JS, Nishimura T, Finch SC 2020. Ciguatera fish poisoning: the risk from an Aotearoa/New Zealand perspective. Toxins 12: 50; doi:10.3390/toxins12010050.
Reference was added.
Reviewer 3 Report
Very nice review that explains clearly the challenges of ciguatera research, diagnosis, management, the potential higher distribution of responsible species due to climate change and anthropogenic influence. It also emphasizes the need to continue with interdisciplinary collaboration in order to decrease the number of ciguatera intoxications.
I only have a couple of suggestions:
In the CTX vectors or in modern testing section, I would mention briefly the long extraction procedures for CTX depending on the matrix.
I also suggest to refer to the review MS of Núñez-Vázquez et al 2019. Ciguatera in Mexico (1984-2013) that reports diverse data from a geographic region were a low number of scientific publications exist.
Other minor corrections are:
Lines 71, 77, 91, 92: Gambierdiscus, Fukuyoa in italics
Line 89. Figures, without ´
Lines 183, 268, 370: microalgae (do not separate)
Line 196: species, eliminate´
Line 215: animals, eliminate ´
Line 260. Separate food web
Line 322. . Toxin identification
Line 492 . Therefore
Author Response
Dear reviewer 3,
Thank you for taking the time and effort to review our article and to provide thoughtful responses and points for consideration. We have added the requested references and below you can find a point by point response addressing each suggestion/comment.
Thank you again and kind regards.
Reviewer 3
I only have a couple of suggestions:
In the CTX vectors or in modern testing section, I would mention briefly the long extraction procedures for CTX depending on the matrix.
We have added a statement for this information and added a reference to the FAO/WHO 2020 report discussing this difficulty.
I also suggest to refer to the review MS of Núñez-Vázquez et al 2019. Ciguatera in Mexico (1984-2013) that reports diverse data from a geographic region were a low number of scientific publications exist.
Thank you for the insight, we have added this reference as well as Mar. Drugs 2020, 18(10), 504;
Other minor corrections are:
Lines 71, 77, 91, 92: Gambierdiscus, Fukuyoa in italics
We have corrected this error.
Line 89. Figures, without ´
We have removed the punctuation.
Lines 183, 268, 370: microalgae (do not separate)
We have removed the error.
Line 196: species, eliminate´
We have removed the punctuation.
Line 215: animals, eliminate ´
We have removed the punctuation.
Line 260. Separate food web
We have added the space. .
Line 322. . Toxin identification
We have added the punctuation.
Line 492 . Therefore
We have changed the punctuation.